# The Optimization of Extraction Process, Antioxidant, Whitening and Antibacterial Effects of Fengdan Peony Flavonoids

**DOI:** 10.3390/molecules27020506

**Published:** 2022-01-14

**Authors:** Jie Lu, Zhiqiang Huang, Yusheng Liu, Huimin Wang, Min Qiu, Yinghui Qu, Wenpeng Yuan

**Affiliations:** Heze branch of Qilu University of Technology (Shandong Academy of Sciences), Biological Engineering Technology Innovation Center of Shandong Province, Heze 274000, China; lujiesunshine@163.com (J.L.); hzqlucky@163.com (Z.H.); liuyusheng666@126.com (Y.L.); whm_hy@sina.com (H.W.); qmqh2018@163.com (M.Q.); QYH9020@163.com (Y.Q.)

**Keywords:** Fengdan peony flavonoid, antioxidant, whitening, antibacterial activity, functional food

## Abstract

Flavonoids have important biological activities, such as anti-inflammatory, antibacterial, antioxidant and whitening, which is a potential functional food raw material. However, the biological activity of Fengdan peony flavonoid is not particularly clear. Therefore, in this study, the peony flavonoid was extracted from Fengdan peony seed meal, and the antioxidant, antibacterial and whitening activities of the peony flavonoid were explored. The optimal extraction conditions were methanol concentration of 90%, solid-to-liquid ratio of 1:35 g:mL, temperature of 55 °C and time of 80 min; under these conditions, the yield of Fengdan peony flavonoid could reach 1.205 ± 0.019% (the ratio of the dry mass of rutin to the dry mass of peony seed meal). The clearance of Fengdan peony total flavonoids to 1,1-diphenyl-2-picrylhydrazyl (DPPH) free radical, hydroxyl radical and 2,2’-azino-bis (3-ethylbenzothiazoline-6-sulfonic acid) (ABTS) free radical could reach 75%, 70% and 97%, respectively. Fengdan peony flavonoid could inhibit the growth of the Gram-positive bacteria. The minimal inhibitory concentrations (MICs) of Fengdan peony flavonoid on *S. aureus*, *B. anthracis*, *B. subtilis* and *C. perfringens* were 0.0293 mg/mL, 0.1172 mg/mL, 0.2344 mg/mL and 7.500 mg/mL, respectively. The inhibition rate of Fengdan peony flavonoid on tyrosinase was 8.53–81.08%. This study intensely illustrated that the antioxidant, whitening and antibacterial activity of Fengdan peony total flavonoids were significant. Fengdan peony total flavonoids have a great possibility of being used as functional food materials.

## 1. Introduction

Peony is one of the economic and ornamental crops, which has been cultivated for more than 1600 years in China [1]. Oil peony refers to the variety of peony that can be used as a raw material for edible oil [2]. As one of the most commonly used oil peonies, Fengdan peony contains dark oval seeds, which has abundant unsaturated fatty acids (>90%) [3]. Peony seed oil has been authenticated as a new resource food by the Chinese government, because of its high proportion of α-linolenic acid in 2011 [4]. Moreover, peony seeds have been certified as a new resource of functional food to enhance human health by the National Health and Family Planning Commission of the People’s Republic of China in 2011 [5]. Fengdan peony seed meal is one of the most important by-products in preparation of peony seed oil, which accounts for approximately 60% to 70% of Fengdan peony seed, and is rich in flavonoids, protein, polysaccharides, polyphenols, paeoniflorin and other active ingredients. However, a large amount of Fengdan peony seed meal is only used as feed or as waste at present [6], which results in the waste of natural resources and environmental pollution [2].

According to literature, flavonoids have important biological activities, such as anti-inflammatory, antibacterial, antioxidant and whitening effects [7,8,9,10], which are considered potential functional food raw materials. For example, soybean isoflavone can protect cells from the damage of free radicals because of its antioxidant properties [11,12,13]. Flavonoid in ginkgo biloba extract has strong antioxidant properties and can directly inhibit free radicals [14]. The flavonoid extract of white guava is a natural antibacterial agent, which can change the microscopic morphology of *E. coli* and *S. aureus* [15].

Unfortunately, as a potential functional food raw material, Fengdan peony flavonoid has not only been involved in industrial production, but its biological activity is also not particularly clear. Therefore, in order to improve the application value of peony flavonoids in the food field as much as possible and reduce the waste of resources, peony flavonoid was extracted from the Fengdan peony seed meal, and the antioxidant, whitening and antibacterial activities of Fengdan peony flavonoid were studied. It is expected to lay a foundation for the development of new functional food raw materials from Fengdan peony.

## 2. Results and Discussion

### 2.1. The Influence of Four Factors to Total Flavonoids Yield

The influence of methanol concentration to total flavonoids yield was shown in Figure 1a. Total flavonoid yield was increased at first, and then decreased with the increase of methanol concentration, and reached the maximum (1.016%, the ratio of the dry mass of rutin to the dry mass of peony seed meal) when methanol concentration was 90%. The reason may be related to the polarity of methanol and the solubility of total flavonoids in peony seed meal [16]. Thus, the optimal methanol concentration was 90%.

The influence of solid-to-liquid ratio to total flavonoids yield was illustrated in Figure 1b. The total flavonoids yield was increased from 0.874% to 1.030% as the solid-to-liquid ratio increased from 1:25 to 1:35. It may be that the contact area between seed meal and methanol solution was increased with the increase of the solid-to-liquid ratio, which led to the effective dissolution of total flavonoids. In the range of 1:35–1:45, total flavonoids yield was decreased from 1.030% to 0.735%. It may be due to the fact that the solubility of non-flavonoids was increased in the system and competed with total flavonoids for solvents [17]. Thus, 1:35 was the best solid-to-liquid ratio.

As shown in Figure 1c, total flavonoids yield was increased at first and then decreased with the increase of temperature. Total flavonoids yield reached the maximum value (1.141%, the ratio of the dry mass of rutin to the dry mass of peony seed meal) at 55 °C. The dissolution rate of total flavonoids in seed meal was accelerated by raising the temperature appropriately. However, a partial total flavonoid was decomposed when it was exposed at high temperature [18], which leads to a decrease in the yield of total flavonoids. Thus, 55 °C was the best temperature.

The influence of extraction time to total flavonoids yield was shown in Figure 1d. Total flavonoids yield was increased rapidly in 0.5 h to 1.5 h. When the extraction time was more than 1.5 h, total flavonoids yield remained unchanged. The reason for this phenomenon may be that the total flavonoids in seed meal were continuously dissolved with the extension of extraction time in 0.5 h to 1.5 h. Furthermore, with the time reaching 1.5 h, all total flavonoids may have been dissolved. Therefore, increasing the time had a slight effect on total flavonoids yield [4,19]. Thus, the best extraction time was 1.5 h.

### 2.2. Optimization for Total Flavonoids Yield

#### 2.2.1. Model Fitting and Data Analysis Using RSM

To further explore the relative importance of the various factors to the total flavonoids yield of Fengdan peony seed meal, the RSM experiment was carried out. RSM is used in the statistical design and data analysis of multifactor experiments to assess the relative significance of several variables, and to find the best conditions for ideal responses [20]. The Box–Behnken design with RSM can fit the linear regression equation, so the experiment can be more precisely analyzed to find the law among the influencing factors [21].

Appendix A showed the relationship between the total flavonoids yield and the test variables. The experimental data were analyzed by multiple regression analysis. The quadratic polynomial regression equation of total flavonoids yield (Y) to solid-to-liquid ratio (A), extraction temperature (B) and extraction time (C) was: $y=1.19+0.01A-0.02B-0.003C+0.02\mathrm{AB}-0.02\mathrm{AC}+0.01\mathrm{BC}-0.03A^{2}-0.04B^{2}—0.02C^{2}.$

The anova of the quadratic polynomial model was demonstrated in Table 1. The F-value of model was 18.57, the *p* value of model was 0.0004 (*p* < 0.01), the fitting degree of the model was 0.1202, and the determination coefficient (R^2^) was 0.9598, which indicated that the model had extremely significant statistical significance. Primary term (B) and secondary term (A^2^, B^2^, C^2^) had extremely significant effects on total flavonoids yield (*p* < 0.01). A, AB and AC had significant effects on total flavonoids yield (*p* < 0.05). C and BC had no significant effect on total flavonoids yield (*p* > 0.05). According to the F-value of the three factors, the effect on total flavonoids yield was extraction temperature > solid-to-liquid ratio > extraction time.

The three-dimensional (3D) response surface diagrams were shown in Figure 2a,c,e. The highest point of the 3D graph was the best condition for interactive factors. The contour slopes of the interaction between A and B (Figure 2b), A and C (Figure 2d) were steeper, while those of A and C (Figure 2f) were smooth. This illustrated that the interaction between A and B, A and C had a great influence on total flavonoids yield, while the interaction between B and C had no significant effect on total flavonoids yield. The consistency between the RSM experiment and regression analysis further proved that the established model was more accurate.

#### 2.2.2. Verification of Optimal Conditions

The theoretical optimum parameters given by the model were solid-to-liquid ratio of 1:36.11, extraction temperature of 53.52 °C, extraction time of 82.65 min, and the theoretical value of total flavonoids yield was 1.195%. Considering the actual operation, the theoretical parameters were adjusted to the solid-to-liquid ratio of 1:35, the extraction temperature of 55 °C, and the extraction time of 80 min. Under these conditions, total flavonoids yield was 1.205 ± 0.019%. Compared with the theoretical forecast value, the relative error was 0.8%. The theoretical value was consistent with the actual result, indicating that the optimization parameters were available. Chen et al. [10] extracted flavonoids from peony using 60% ethanol-aqueous solution and the yield was 1.34%, which were consistent with this paper. RSM was widely used to optimize the optimal extraction process of flavonoids from plants, such as Aurantii Fructus [22], Pueraria [23], and Curcuma Zedoaria leaves [24]. Compared with the orthogonal test, RSM had the advantages of accurate predictability, higher accuracy, and more easily analyzing influence factors [25].

### 2.3. Antioxidant Activity of Fengdan Peony Flavonoid

Flavonoids are natural antioxidants which can effectively clean free radicals in the body. The antioxidant capacity of the flavonoids is due to the fact that flavonoids can provide hydrogen atoms to free radicals and convert themselves into phenolic free radicals. The transfer rate of the automatic oxidation chain reaction can be slowed down by the stability of phenolic free radicals, thus inhibiting the further oxidation of lipids [26].

The clearance of total flavonoids to DPPH free radical was shown in Figure 3a. In the concentration range of 0.1 to 1.0 mg/mL, the clearance of total flavonoids and VC to DPPH free radicals was positively correlated with their concentration. In the range of 0.1 to 0.5 mg/mL, the clearance of VC was increased with its concentration. When the concentration was greater than 0.5 mg/mL, the clearance of VC reached 98% and tended to be stable. In the range of 0.1 to 0.8 mg/mL, the clearance of total flavonoids was increased with the increase of concentration. When the concentration was greater than 0.8 mg/mL, the clearance of total flavonoids stabilized to about 75%. Oancea et al. [27] reported that the clearance of peony crude extract to DPPH free radical was 81%; the clearance was similar to the results of this study. Fengdan peony total flavonoids had a certain clearing capacity to DPPH free radicals, but the capacity was slightly lower than VC. The reaction between antioxidant and DPPH free radical is realized by hydrogen atom and electron transfer mechanism [28]. DPPH free radical is purple in ethanol solution. When the antioxidant scavenges DPPH free radical to a stable state through hydrogen supply, its color turns yellow and has strong absorption at 517 nm [29]. In this paper, the scavenging effect of peony flavonoids on DPPH was significantly enhanced with the increase of flavonoid concentration, which indicated that the peony flavonoids were effective antioxidants.

The clearance of total flavonoids to hydroxyl radical was illustrated in Figure 3b. In the concentration of 0.1 to 1.0 mg/mL, the clearance of total flavonoids to hydroxyl radical was lower than VC. At 0.4 mg/mL, the clearance of VC basically reached the maximum, about 97%. The scavenging rate of total flavonoids on hydroxyl radicals was increased with the raise of concentration, and reached the maximum value (70%) when the concentration was higher than 0.8 mg/mL. Hydroxyl radical is quite active and could react quickly with any biomolecule, which causes great harm to organs or tissues [30]. Hydroxyl radical is the most toxic reactive oxygen species to cells [31], which accelerates cell apoptosis by enhancing the oxidation of the body [32]. Yang et al. [33] found that the scavenging effect of peony seed oil on hydroxyl radical was concentration-dependent, and the clearance of hydroxyl radical was up to 92% in the concentration range of 0.1–0.5 mg/mL. Compared with this paper, the clearance of peony seed meal flavonoids was lower than that of peony seed oil, but as a by-product in the production of peony seed oil, the flavonoids extracted from peony seed meal also had obvious scavenging effect on hydroxyl radical at lower concentration. Therefore, peony flavonoids could be considered as a functional food raw material with the ability of a scavenging hydroxyl radical. On the one hand, it had a better protective effect on the human antioxidant defense system, and on the other hand, it could also improve the comprehensive utilization of peony seeds.

Figure 3c showed that peony total flavonoids had a strong ABTS radical cleaning ability within the experimental concentration range. The clearance increased in a significant concentration-dependent manner with the increase of total flavonoids concentration. Similarly, Bai et al. [34] also found that the antioxidant activities determined by ABTS assays showed significant correlations with peony total flavonoid content. When the concentration reached up to 0.7 mg/mL, the clearing ability of peony total flavonoids to ABTS free radicals were close to VC. ABTS free radicals have a maximum absorption at 734 nm, when a substance is added to ABTS radicals solution and its absorbance decreases at 734 nm, which indicated that the substance had the ability to clean ABTS radicals and belongs to an antioxidant [35]. In our study, peony flavonoids could effectively scavenge ABTS free radicals. Therefore, peony flavonoids can be added to health products or functional foods as plant antioxidants to enhance the antioxidant effect of food and delay aging.

### 2.4. Whitening Activity of Fengdan Peony Flavonoid

As shown in Figure 4, Fengdan peony flavonoid (0.03–1.00 mg/mL) produced 8.53–81.08% inhibition of tyrosinase, which exhibited tyrosinase inhibition dose-dependently. The IC_50_ was 0.24 mg/mL. Lin et al. [36] found that the inhibition rate of tyrosinase was about 75% when the concentration of ethanol extract from peony was 1 mg/mL. The inhibition rate of tyrosinase was close to the results of this paper. Melanin is the key substance to inhibit whitening [37]. Tyrosinase is a crucial enzyme in the synthesis of melanin, which is used as an index for screening cosmetic and whitening functional foods [38]. Tyrosinase inhibitors are essential ingredients in most cosmetics or whitening foods [39]. In this study, the peony flavonoid had an obvious inhibitory effect on tyrosinase activity. Therefore, it was feasible and valid to add peony flavonoids to cosmetics or whitening foods as whitening materials.

### 2.5. Antibacterial Activity 

The antibacterial properties make great significance for the development and marketization for health or functional food [40]. Thus, the antibacterial ability of the Fengdan peony flavonoid was explored by using the inhibition zone experiment and MIC. The results of the inhibition zone experiment were showed in Appendix A and Figure 5. No matter whether the concentration of peony total flavonoids was 30 mg/mL or 500 mg/mL, all 4 kinds of Gram-positive bacteria had inhibition zones, while three kinds of Gram-negative bacteria and two kinds of fungi had no inhibition zone. It was indicated that peony total flavonoids had a certain antibacterial effect on Gram-positive bacteria, but had no antibacterial effect on Gram-negative bacteria and fungi. Yan et al. [41] found that the extract of peony seeds showed strong activity on Gram-positive bacteria and relatively weak inhibition on fungi. This was similar to our study. The inhibition zones of Gram-positive bacteria from large to small were *C. perfringens* > *S. aureus* > *B. anthracis* > *B. subtilis* (Table 2). As shown in Appendix A, the MIC of peony total flavonoids to *S. aureus* was 0.0293 mg/mL, to *B. anthracis* was 0.1172 mg/mL, to *B. subtilis* was 0.2344 mg/mL, and to *C. perfringens* was 7.500 mg/mL. Lipophilicity is the key reason for flavonoids inhibiting Gram-positive bacteria, and the cell membrane is the main part of flavonoids acting on Gram-positive bacteria [42]. Flavonoids inhibit the biofilm formation of *S. aureus* overexpressing efflux protein genes [43]. This may be one of the reasons why peony flavonoids had bacteriostatic effects on Gram-positive bacteria but had no bacteriostatic effect on Gram-negative bacteria in our study. Therefore, peony flavonoids were natural antibacterial agents, which could be added to detergents, hand sanitizers, and other disinfection products. In addition, as an antibacterial ingredient, peony flavonoids also had potential application value in feed. For example, *C. perfringens* was considered to be the pathogen of most chicken necrotizing enteritis [44], so adding peony total flavonoids to chicken feed may inhibit its reproduction in the small intestine, to a certain extent.

## 3. Materials and Methods

### 3.1. Material and Reagents

The skimmed Fengdan peony seed meal was purchased from Guyu Peony Biotechnology Co., Ltd. (Heze, China). The seed meal was ground into powder and then passed through a 40-mesh sieve [45]. The seed meal powder was collected and stored at 4 °C for future use.

Rutin standard, Vitamin C (VC), Methanol, Anhydrous ethanol Sodium nitrite (NaNO_2_), aluminum nitrate (Al(NO_3_)_3_) and Sodium hydroxide (NaOH) were supplied by China Pharmaceutical Group Co., Ltd. (Beijing, China); Hydroxyl radical, DPPH free radical and ABTS free radical scavenging capacity kit were purchased from Shanghai Tongwei Industrial Co., Ltd. (Shanghai, China); *E. coli*, *S. typhimurium*, *P. aeruginosa*, *S. aureus*, *B. subtilis*, *B. anthracis*, *C. perfringens*, *C. albicans* and *A. niger* were purchased from Beina chuanglian Biotechnology Research Institute (Xinyang, China); Fluid thioglycollate medium and Columbia agar medium were used for the culture of *C. perfringens*; Sabouraud dextrose broth medium and Potato dextrose agar medium were used for the culture of *C. albicans* and *A. niger*; Nutritional agar medium and nutritional broth medium were used for the culture of other bacteria; all mediums were purchased from Qingdao Rishui Co., Ltd. (Qingdao, China) Anaerobic airbag and anaerobic culture bag for anaerobic culture of *C. perfringens* were purchased from Qingdao Rishui Co., Ltd. (Qingdao, China).

### 3.2. Extraction of Total Flavonoids 

Moreover, 15.00 g skimmed Fengdan peony seed meal was added into a 1000 mL beaker. Then, it was placed in a digital display thermostatic water bath (HH-S_4_, Jiangsu, China) with an electric blender (OES-60M, Wenzhou, China, 220 r/min) and extracted under the following conditions: methanol concentration was 90%, solid-to-liquid ratio was 1:35 g:mL, extraction temperature was 55 °C, and extraction time was 80 min. Upon extraction, the filtrate (500 mL) of total flavonoids was collected by vacuum filter (SHZ-DIII, Gongyi, China).

### 3.3. Determination of Fengdan Peony Total Flavonoids Yield

Total flavonoids yield was measured by NaNO_2_-Al(NO_3_)_3_-NaOH method [46]. First, 0.1 mg/mL rutin solution was prepared by 80% (volume fraction) methanol solution. Then, different volumes (0.00, 1.00, 2.00, 3.00, 4.00, 5.00, 6.00 mL) of 0.1 mg/mL rutin solution and 1.00 mL filtrate of total flavonoids were added into 7 tubes. The solution of each tube was replenished to 6 mL with 80% (volume fraction) methanol. Next, 0.3 mL of 5% (mass fraction) NaNO_2_, 0.3 mL of 10% (mass fraction) Al(NO_3_)_3_ and 3.0 mL of 4% NaOH were added into the tube, and stood at room temperature for 7 min, 7 min and 15 min, respectively [47]. Finally, the absorbance was measured by ultraviolet spectrophotometry (T2602S, Shanghai, China) at 510 nm.

As shown in Figure 6, the standard curve was drawn with the concentration of rutin as horizontal coordinate and the absorbance as ordinate. The regression equation was $y=10.122x-0.0022,R^{2}=0.9996$. Furthermore, the absorbance of peony total flavonoids filtrate was brought into the regression equation to obtain its concentration. The yield was calculated by the following formula:(1)$$\mathrm{Total}\;\mathrm{flavonoids}\;\mathrm{yield}=\frac{C\times V}{M\times 1000}\times 100\%\;$$
where the C was the concentration of 1.00 mL filtrate of total flavonoids, V was the total volume of extraction solution, and M was the dry mass of the peony seed meal.

### 3.4. Single-Factor Experiment

The total flavonoids yield of Fengdan peony seed meal was optimized by using single-factor experiment. It could reflect the effect of four factors on total flavonoids yield. This experiment was investigated by the single variable method. Factors and variables of the experiment were shown in Table 3.

### 3.5. RSM Experiment

The single-factor experiment provided three variables of each factor for the RSM experiment. In this study, on the basis of single-factor experiment, methanol concentration was selected to be 90%; solid-to-liquid ratio, extraction temperature, and extraction time were used as arguments. The peony total flavonoids yield was used as the response value. Based on the principle of the Box–Behnken test in Design-Expert software 12.0, the RSM with three factors and three levels was designed to optimize the extraction conditions of total flavonoids from peony seed meal. The whole experimental design was composed of 17 experimental points (Appendix A). Five replications (13–17) were performed in the design center to estimate a pure error. The factors and levels of RSM were demonstrated in Table 4.

### 3.6. Preparation of Fengdan Peony Total Flavonoids Powder

After the optimum extraction process was determined by RSM, the Fengdan peony total flavonoids filtrate was extracted with the best extraction process, then the peony total flavonoids powder was obtained after the filtrate was concentrated by rotary evaporator (20 mL/min, 25 W, R-1001VN, Zhengzhou, China) and dried by freeze dryer (−80 °C, 24 h, SCIENTZ-12N, Ningbo, China). The powder would be used in antioxidant, whitening and antibacterial experiments.

### 3.7. Antioxidant Activity Evaluation

Fengdan peony total flavonoids solution and VC solution with different mass concentrations (0.1, 0.2, 0.3, 0.4, 0.5, 0.6, 0.7, 0.8, 0.9, 1.0 mg/mL) were prepared with 80% ethanol, which will be used to determine the clearance of total flavonoids to three free radicals. 

#### 3.7.1. DPPH Free Radical Assay

The clearance of total flavonoids to DPPH free radical was determined as described by Pham, D. C., with minor modifications [48]. Briefly, 150 μL of 0.2 mM DPPH in anhydrous ethanol was added to 150 μL of sample at different concentrations (0.1–1.0 mg/mL). The mixture was shaken and incubated for 30 min at room temperature in the dark. The 200 μL mixture was added into a 96-well plate, and the absorbance was measured at 517 nm by enzyme-labeled instrument (SpectraMax 190, Shanghai, China). VC was used as the control substance. The clearance of DPPH was calculated by the following formula.
(2)$$\mathrm{The}\;\mathrm{clearance}\;\mathrm{of}\;\mathrm{DPPH}\;\%=\left[1-\left(\frac{A_{i}-A_{j}}{A_{0}}\right)\times 100\;\right]\%$$
where Ai was the absorbance of 150 μL sample + 150 μL DPPH; A_j_ was the absorbance of 150 μL sample + 150 μL anhydrous ethanol; A_0_ was the absorbance of 150 μL DPPH + 150 μL anhydrous ethanol.

#### 3.7.2. Hydroxyl Radical Assay

The clearance of total flavonoids to hydroxyl radical was determined as described by Zhou, J., with some modifications [49]. Briefly, 50 μL sample solutions of different concentrations (0.1–1.0 mg/mL), 50 μL of 9.0 mM salicylic acid-ethanol solution, 50 μL of 9.0 mM FeSO_4_ solution and 200 μL of distilled water were mixed in a tube. Then, 50 μL of 8.8 mM H_2_O_2_ was added into the mixture above, and the absorbance was measured at 510 nm. The clearance of hydroxyl radical was calculated by the following formula.
(3)$$\mathrm{The}\;\mathrm{clearance}\;\mathrm{of}\;\mathrm{hydroxyl}\;\mathrm{radical}\;\%=\left[1-\left(\frac{A_{i}-A_{j}}{A_{0}}\right)\times 100\;\right]\%$$
where A_i_ was the absorbance of the sample; A_j_ is the absorbance of deionized water instead of H_2_O_2_; A_0_ was the absorbance of deionized water instead of sample. 

#### 3.7.3. ABTS Free Radical Assay

The clearance of total flavonoids to ABTS free radical was determined as described by Gong, J. with some modifications [50]. Briefly, 1 mL of 7 mM ABTS solution and 1 mL of 2.45 mM potassium persulfate solution were mixed in a tube for 12 h at room temperature in the dark. The mixture was diluted 40 times with anhydrous ethanol. Then, 190 μL of mixture and 10 μL of samples at different concentrations (0.1–1.0 mg/mL) were mixed in a tube for 6 min at room temperature in the dark; the absorbance was measured at at 734 nm. The clearance of ABTS free radical was calculated by the following formula.
(4)$$\mathrm{The}\;\mathrm{clearance}\;\mathrm{of}\;\mathrm{ABTS}\;\%=\left[1-\left(\frac{A_{i}-A_{j}}{A_{0}}\right)\times 100\;\right]\%$$
where Ai was the absorbance of 10 μL sample + 190 μL mixture; A_j_ was the absorbance of 10 μL sample + 190 μL anhydrous ethanol; A_0_ was the absorbance of 10 μL anhydrous ethanol + 190 μL mixture.

### 3.8. Whitening Activity Evaluation

The tyrosinase inhibition assay was determined as described by Chen Q with minor modifications [38]. Briefly, 40 μL of 5 mmol/L tyrosine dissolved in 80 μL  of 1/15 phosphate buffers (pH 6.8) was mixed with 40  μL Fengdan peony total flavonoids solution at 37 °C for 15 min. Next, 40 μL of tyrosinase (300 IU/mL) was added to initiate the reaction. The assay mixture was incubated at 37 °C for 10 min. The absorbance was measured by an enzyme-labeled instrument (SpectraMax 190, Shanghai, China) at 482 nm. At the same time, the blank group and the control group were set up. In the blank group, tyrosinase was not added and the phosphate buffer was used to make up the volume. In the control group, the flavonoid solution was not added, and the phosphate buffer was used to make up the volume. The inhibitory percentage of the tyrosinase activity was calculated by the following formula:(5)$$\mathrm{Inhibition}\;\mathrm{rate}\;\left(\%\right)=\left(1-\frac{A_{S}-A_{B}}{A_{C}-A_{B}}\right)\times 100\%$$
where A_B_ was the absorbance of blank group at 482 nm, A_C_ was the absorbance of control group at 482 nm, and A_S_ was the absorbance of sample at 482 nm.

### 3.9. Antibacterial Activity Experiment

Then, 400 μL of sterile water was added into the bacterial freeze-dry tube. After being fully mixed, 200 μL of bacteria liquid was evenly spread agar plate. It was incubated in a biochemical incubator (LRH-250, 650 W, Changzhou, China) at 37 °C for 24 h to activate the bacteria. Then, the single colony was picked in agar plate with a disposable inoculation loop (65-0001, Shandong, China) and inoculated in the liquid medium. The medium was placed in an intelligent constant temperature culture oscillator (HNYC-202T, 1800W, Tianjin, China) at 37 °C, 120 r/min for 24 h. The concentration of bacterial suspension was observed and counted by hemocytometer (Q401, Shanghai, China), and then adjusted to 10^6^ CFU/mL [51].

The inhibition zone experiment was operated by the punching method. First, Fengdan peony total flavonoids solutions of different concentrations (30 mg/mL and 500 mg/mL) were prepared with 80% methanol. Then, the 80 μL of bacteria suspension was spread evenly on the sterile agar medium. Next, 4 holes were punched in the medium by a puncher with a diameter of 7 mm (A0358, Guangzhou, China). Three of the holes were added to 100 μL of total flavonoids solution, and the other was added to the 100 μL of 80% methanol as a control. Final, it was cultured in a biochemical incubator at 37 °C for 24 h. The diameter of the clear inhibition zone around each hole was measured by using a vernier caliper (SWB-227-150, Guangzhou, China).

The MIC experiment was performed according to the following steps. The bacterial growth system containing different concentrations of total flavonoids solution (0.0037, 0.0073, 0.0146, 0.0293, 0.0586, 0.1172, 0.2344, 0.4688, 0.9375, 1.8750, 3.7500, 7.5000, 15.0000 mg/mL) was prepared by a double gradient dilution method [52]. In addition, adding bacterial solution without total flavonoids solution was used as growth control group, and total flavonoids solution without bacterial solution was used as a blank control group. They were cultured in intelligent constant temperature culture oscillator (HNYC-202T, 1800 W, Tianjin, China) at 37 °C, 120 rpm/min for 24 h. The maximum diluted concentration of total flavonoids without bacterial growth was the MIC of total flavonoids on the bacteria [53].

### 3.10. Statistical Analysis

Design-Expert V12.0.3.0 (Stat-Ease Inc., Minneapolis, MN, USA) was used to calculate the coefficients of the quadratic polynomial model and the optimization. F-values and *p*-values were used to check the accuracy of the polynomial model equation, and *p*-values less than 0.05 (*p* < 0.05) were considered as statistically significant. All experiments were measured three times, and the data values were expressed as means ± standard deviation (SD).

## 4. Conclusions

In this study, the optimum extraction conditions of Fengdan peony total flavonoids were methanol concentration of 90%, solid-to-liquid ratio of 1:35, extraction temperature of 55 °C, extraction time of 80 min, and the total flavonoids yield was 1.204%. We believe that the extraction process is suitable for industrial production. The Fengdan peony flavonoids demonstrated excellent antioxidant effects, the clearance of total flavonoids to DPPH free radical was 75%, to hydroxyl radical was 70%, and to ABTS free radical was 97%. In addition, the Fengdan peony flavonoids had a certain whitening effect, the inhibition rate of Fengdan peony flavonoids to tyrosinase exhibited dose-dependently and the IC_50_ could reach to 0.24 mg/mL. Moreover, the Fengdan peony total flavonoids could inhibit the growth of Gram-positive bacteria. All the results provided a reliable data support for the application of Fengdan peony total flavonoids in health product and functional food raw materials, and laid a foundation for the industrial production of Fengdan peony total flavonoids.

## Figures and Tables

**Figure 1 molecules-27-00506-f001:**
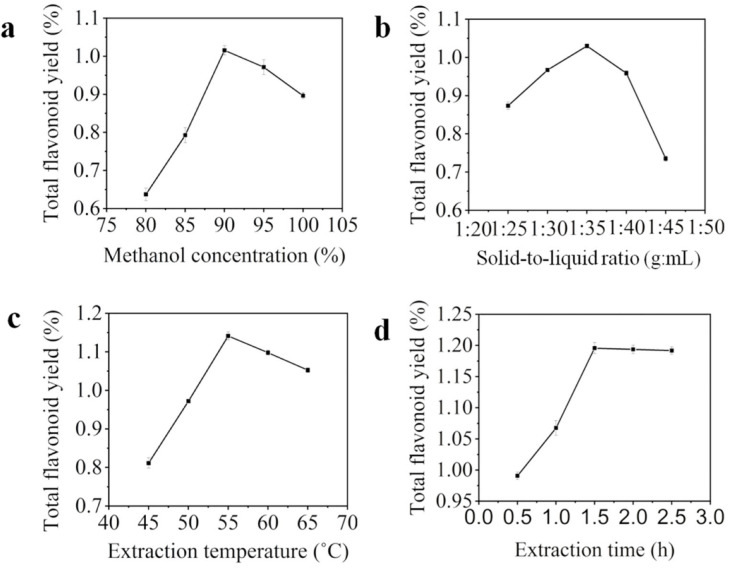
The influence of methanol concentration (**a**), solid-to-liquid ratio (**b**), extraction temperature (**c**) and extraction time (**d**) to total flavonoids yields, respectively.

**Figure 2 molecules-27-00506-f002:**
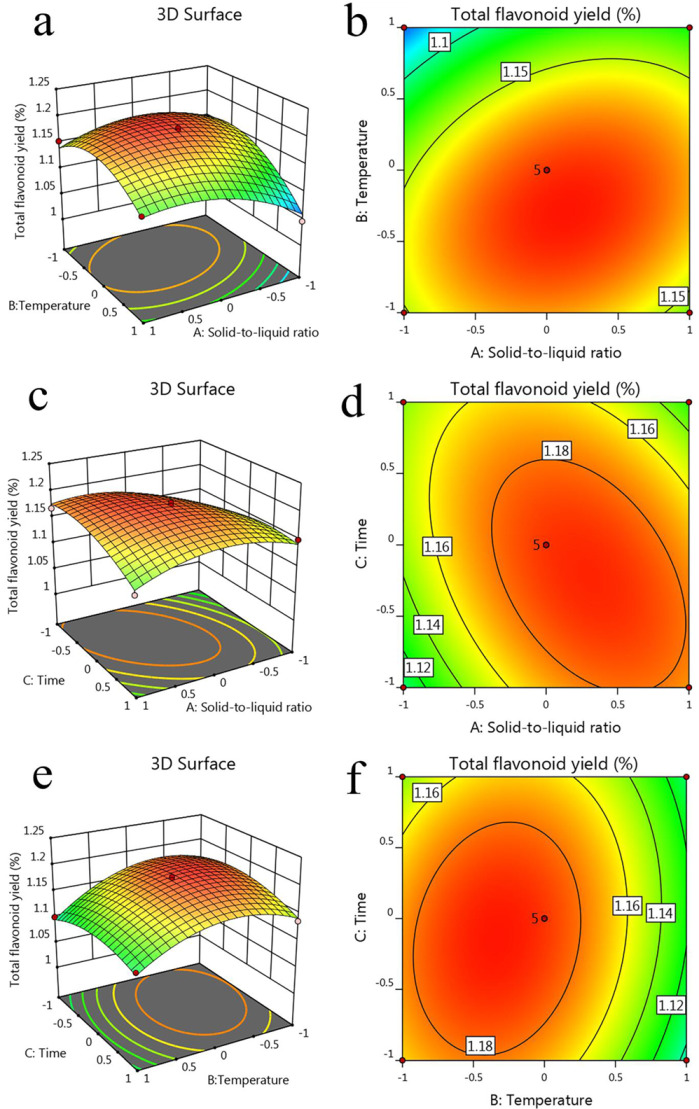
The diagrams of 3D surface for AB (**a**), AC (**c**) and BC (**e**) to total flavonoids yield. The diagrams of contour for AB (**b**), AC (**d**) and BC (**f**) to total flavonoids yield.

**Figure 3 molecules-27-00506-f003:**
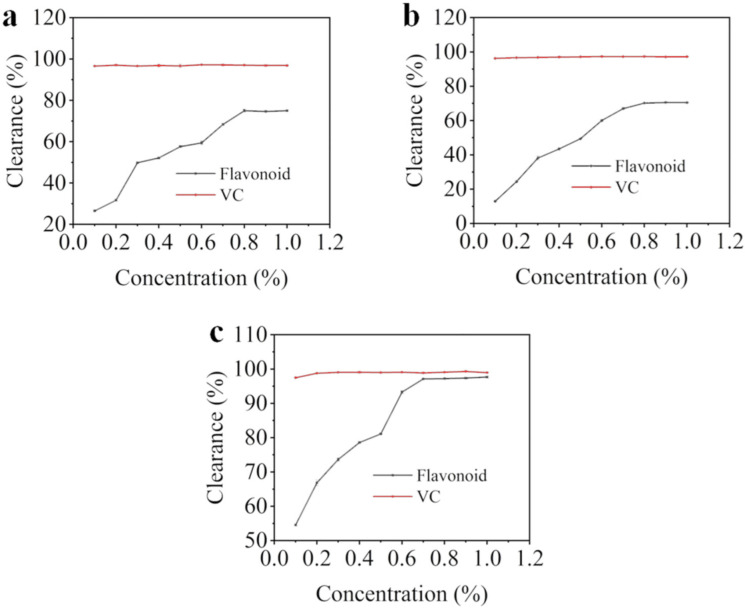
The clearance of total flavonoids to DPPH free radical (**a**), hydroxyl radical (**b**) and ABTS free radical (**c**).

**Figure 4 molecules-27-00506-f004:**
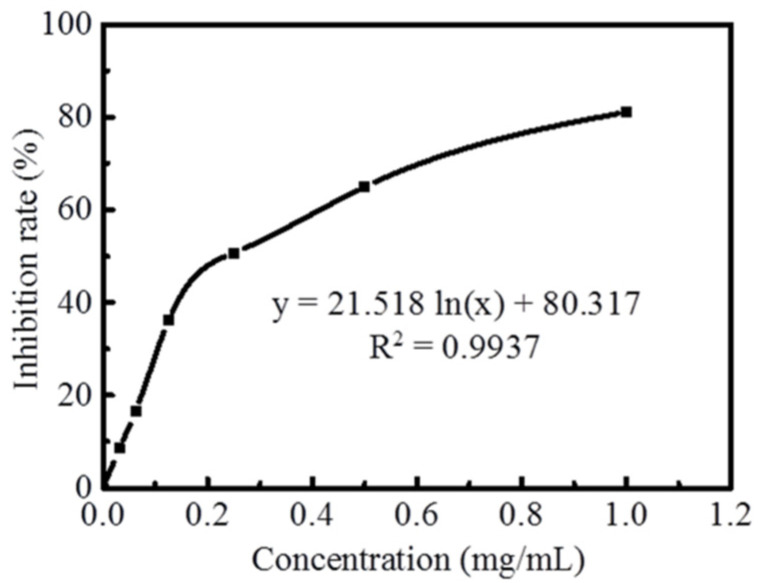
The inhibition rate of total flavonoids to tyrosinase.

**Figure 5 molecules-27-00506-f005:**
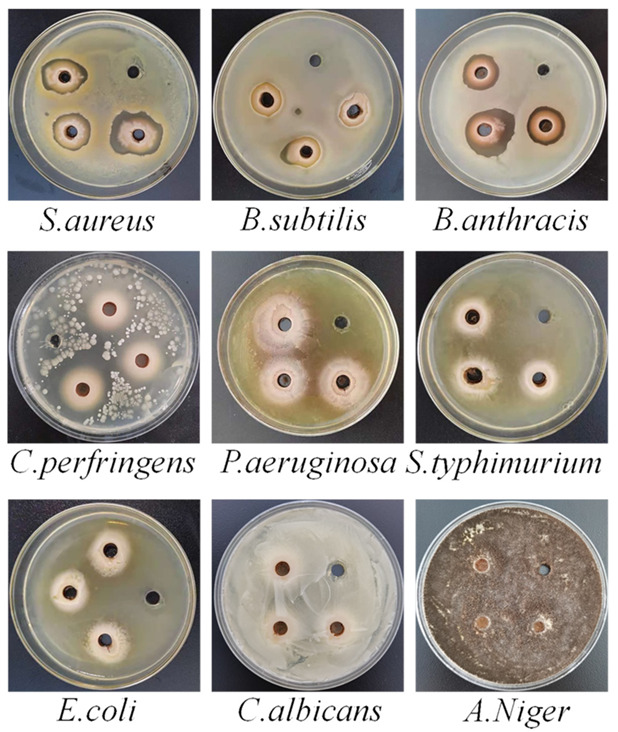
Photographs of inhibition zone experiment for 500 mg/mL peony total flavonoids to 9 kinds of bacteria.

**Figure 6 molecules-27-00506-f006:**
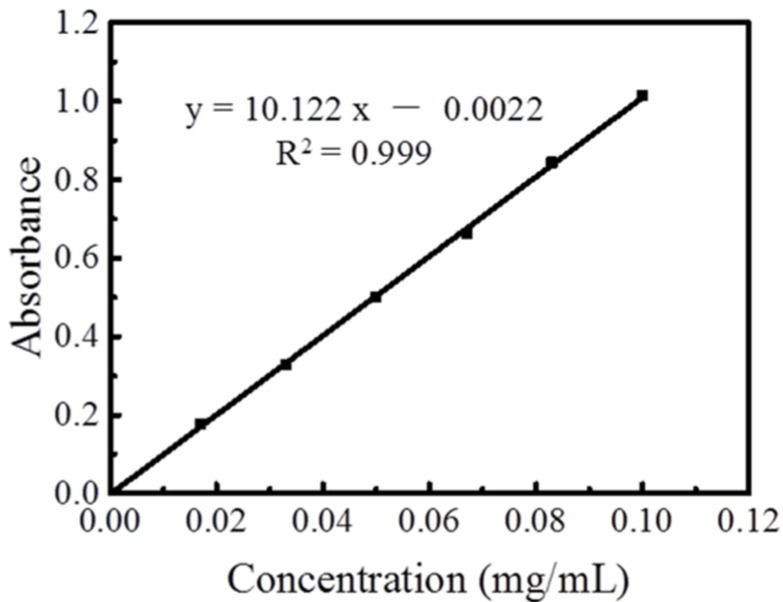
Rutin standard curve.

**Table 1 molecules-27-00506-t001:** The anova of quadratic polynomial model.

Source	Sum of Squares	df	Mean Square	F-Value	*p* Value	Significant
Model	0.0249	9	0.0028	18.57	0.0004	**
A	0.0015	1	0.0015	10.16	0.0153	*
B	0.0055	1	0.0055	37.03	0.0005	**
C	0.0001	1	0.0001	0.4836	0.5092	
AB	0.0013	1	0.0013	8.70	0.0214	*
AC	0.0018	1	0.0018	11.85	0.0108	*
BC	0.0003	1	0.0003	2.18	0.1837	
A ^2^	0.0040	1	0.0040	26.65	0.0013	**
B ^2^	0.0071	1	0.0071	48.00	0.0002	**
C ^2^	0.0019	1	0.0019	12.71	0.0092	**
Residual	0.0010	7	0.0001			
Lack of fit	0.0008	3	0.0003	3.68	0.1202	
Pure Error	0.0003	4	0.0001			
Cor Total	0.0259	16				
R^2^ = 0.9598						

*: *p* <0.05; **: *p* < 0.01.

**Table 2 molecules-27-00506-t002:** Diameter of inhibition zone.

Bacteria	Average Diameter of Inhibition Zone (cm)	
	30 mg/mL	500 mg/mL
*B. subtilis*	1.35 ± 0.18	1.83 ± 0.06
*B. anthracis*	1.95 ± 0.05	2.20 ± 0.26
*S. aureus*	2.27 ± 0.26	2.65 ± 0.33
*C. perfringens*	2.63 ± 0.06	2.97 ± 0.06
*E. coli*	-	-
*S. typhimurium*	-	-
*P. aeruginosa*	-	-
*C. albicans*	-	-
*A. niger*	-	-
80% methanol	-	-

**Table 3 molecules-27-00506-t003:** The factors and variables of single-factor experiment.

Factors	Variables
Methanol concentration (%)	80; 85; 90; 95; 100
Solid-to-liquid ratio (g/mL)	1:25; 1:30; 1:35; 1:40; 1:45
Extraction temperature (°C)	45; 50; 55; 60; 65
Extraction time (h)	0.5; 1; 1.5; 2; 2.5

**Table 4 molecules-27-00506-t004:** The factors and levels of RSM.

Factors	Levels		
	−1	0	1
A Solid-to-liquid ratio (g/mL)	1:30	1:35	1:40
B Extraction temperature (°C)	50	55	60
C extraction time (h)	1	1.5	2

## Data Availability

All available data are contained within the article.

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
