# Peer review of "The Optimization of Extraction Process, Antioxidant, Whitening and Antibacterial Effects of Fengdan Peony Flavonoids"

_molecules, 2022, doi:10.3390/molecules27020506_

Round 1

Reviewer 1 Report

This work is very interesting and deals with a very important topic, which is ecology and the use of industrial waste. However, a scientific article should not be a subjective opinion, therefore the sentence below should be changed; The methodology needs to be refined. However, the experiment was well planned. The work lacks discussion, comparing the results with other authors. This is the most important part of the article.

However, below I am sending a few considerations:

  • line 54 - We firmly believe that the garbage is a misplaced resource for humans. Of course, this is very important, but the value of the manuscript will increase when the authors do not use phrases that speak strictly of their views.
  • For the extraction it seems to me that it is necessary to provide parameters and not just to cite the previous work.
  • the concentration of routine at the calibration curve should be determined. They should not be given ml and stock. It would be much better to specify the amount of rutin in mg / ml
  • The statistical program and selected tests were not given, which is necessary.
  • In the work, the authors mention regression, but in Figure 2 there is no statistics.

Reviewer 2 Report

I think that the manuscript entitled “The optimization of extraction process, antioxidant, whitening and antibacterial effects of Fengdan peony flavonoids " deserves publication in Molecules after major revision. The research is of a practical nature, which the authors themselves emphasized several times. However, in my opinion, it requires thorough changes, especially the discussion of the results, which is practically non-existent. I miss a comparison of how extracts, polyphenols, flavonoids obtained from other plants, side products in the agri-food industry influenced the parameters assessed by you.

Line 17: please complete the information on the flavonoid “%” in dry mass, wet mass, extract, etc.

Line 78-82 : please complete the information so that the experience can be recreated (extraction temperatures, solid-to-liquid ratio, methanol concentrations and extraction time)

Line 78: please complete the information on the water bath (name, type, city, country…)

Line 78: please complete the information on the blender (name, type, city, country…)

Line 82: please complete the information on the filtrate

Line 84-91: please standardize the concentration „%” or “M”

Line 100: please check quality? not quantity?

Line 102: pleas change “Figure.1” into “Figure 1.”

Line 124: please complete the information by concentration and freeze-drying (including: parameters, device, name, type, city, country…)

Line 130: please complete the information on the antioxidant activity evaluation (including: parameters, device, name, type, city, country…)

Line 138: please complete the information by the by enzyme-labeled instrument (including: name, type, city, country…)

Line 152: wrong entry „rpm/min”

Line 148-170: please complete the information on the antibacterial activity experiment (including: parameters, device, name, type, city, country…)

Line 173-174: please complete the information on the statistical analysis (including: program, tests, n, p-value, etc.)

Line 176, 189: please complete the information on the “%” of what

Line 200: please change “[28,4]” into “[4,28]”

Figure 2b: please change “(g/mL)” into “(g:mL)”

Line 256: please change “517nm” into “517 nm”

Line 276, 277: please change “734nm” into “734 nm”

Line 248: please, write a discussion of the results obtained

Line 290: please, write a discussion of the results obtained

Figure 6: please write Latin names in italics

Line 303: please, write a discussion of the results obtained

Round 2

Reviewer 2 Report

I think that the re-submitted manuscript entitled “The optimization of extraction process, antioxidant, whitening and antibacterial effects of Fengdan peony flavonoids " deserves publication in Molecules after minor revision.

Line 87: please complete the information on the filtrate, kind, type of filter

Line 95: please change “1 mol/L NaOH” into „%”

Please complete the information on the spectrophotometr to the measured antioxidant activity evaluation (including: device, name, type, city, country…)

Discussion of the results obtained. That is, a comparison of the obtained results with the results of other authors researching plants i.g. peony.
